# Profiling the Physical Performance of Young Boxers with Unsupervised Machine Learning: A Cross-Sectional Study

**DOI:** 10.3390/sports11070131

**Published:** 2023-07-07

**Authors:** Rodrigo Merlo, Ángel Rodríguez-Chávez, Pedro E. Gómez-Castañeda, Andrés Rojas-Jaramillo, Jorge L. Petro, Richard B. Kreider, Diego A. Bonilla

**Affiliations:** 1Research Division, Dynamical Business & Science Society—DBSS International SAS, Leon 37530, Mexico; jangelcoach@gmail.com; 2Colegio Profesional de Licenciados en Entrenamiento Deportivo (CPLED), Mexico City 03650, Mexico; pedro.gomez@cpled.com.mx; 3Escuela Nacional de Entrenadores Deportivos, Comisión Nacional de Cultura Física y Deporte, Mexico City 08400, Mexico; 4Research Division, Dynamical Business & Science Society—DBSS International SAS, Bogotá 110311, Colombia; arojasj@unal.edu.co (A.R.-J.); jlpetro@dbss.pro (J.L.P.); dabonilla@dbss.pro (D.A.B.); 5Grupo de Investigación CINDA, Instituto Departamental de Deportes de Antioquia (INDEPORTES), Medellín 050034, Colombia; 6Research Group in Physical Activity, Sports and Health Sciences (GICAFS), Universidad de Córdoba, Montería 230002, Colombia; 7Exercise & Sport Nutrition Lab, Human Clinical Research Facility, Texas A&M University, College Station, TX 77843, USA; 8Research Group in Biochemistry and Molecular Biology, Universidad Distrital Francisco José de Caldas, Bogotá 110311, Colombia

**Keywords:** boxing, strength, physical assessment, profiling, machine learning

## Abstract

Mexico City is the location with the largest number of boxers in Mexico; in fact, it is the first city in the country to open a Technological Baccalaureate in Education and Sports Promotion with a pugilism orientation. This cross-sectional study aimed to determine the physical–functional profile of applicants for admission to the baccalaureate in sports. A total of 227 young athletes (44F; 183M; 15.65 (1.79) years; 63.66 (14.98) kg; >3 years of boxing experience) participated in this study. Body mass (BM), maximal isometric handgrip (HG) strength, the height of the countermovement jump (CMJ), the velocity of straight boxing punches (PV), and the rear hand punch impact force (PIF) were measured. The young boxers were profiled using unsupervised machine learning algorithms, and the probability of superiority (ρ) was calculated as the effect size of the differences. K-Medoids clustering resulted in two sex-independent significantly different groups: Profile 1 (*n* = 118) and Profile 2 (*n* = 109). Except for BM, Profile 2 was statistically higher (*p* < 0.001) with a clear distinction in terms of superiority on PIF (ρ = 0.118), the PIF-to-BM ratio (ρ = 0.017), the PIF-to-HG ratio (ρ = 0.079) and the PIF-to-BM+HG ratio (ρ = 0.008). In general, strength levels explained most of the data variation; therefore, it is reasonable to recommend the implementation of tests aimed at assessing the levels of isometric and applied strength in boxing gestures. The identification of these physical–functional profiles might help to differentiate training programs during sports specialization of young boxing athletes.

## 1. Introduction

Boxing is a combat sport between two fighters with their hands sheathed in special gloves, pure acyclic, and in accordance with certain rules [1]. Pugilism or “The Noble Art”, as boxing is known, is one of the oldest combat sports in all human culture [2]. Boxing has fixed intervals of action (3 min fighting) and recovery (1 min resting); therefore, it presents a predominance of the oxidative energy system with important contributions of the phosphagen and glycolysis pathways when performing applied strength and agility gestures [3]. For example, explosive force in the boxing punch is a determining factor for victory by knockout, and it is considered one of the key performance indicators in amateur boxing [4,5,6]. Similarly, the velocity of boxing punches is also associated with increased athletic performance; it has been recorded that a boxing punch has an average fist velocity between 8.9 and 11.5 m·s^−1^ [7,8]. This parameter varies depending on whether the acceleration is measured with a linear encoder, accelerometer, or photometry [9].

Regarding the relationship between strength and technique in boxing, some authors consider that boxers must have optimal development of their muscular strength and power to effectively manage the physical and technical–tactical requirements of the boxing fight [5,10,11,12,13]. Success in boxing has been associated with a high level of physical fitness [14]. For example, it has been reported that the lower limbs contribute 39% of the force achieved in the boxing punch [15]. Since the force generated by the upper and lower extremities is a key determinant of the punching force [2] the evaluation of whole-body physical fitness is important for the profiling of boxers. To accurately establish the physical profile of boxers, we believe that it is necessary not only to use conventional physical tests (e.g., VO_2max_, applied force of lower limbs) but also to integrate specific variables of this sport discipline (e.g., punch velocity (PV) and punch impact force (PIF)).

In line with the aims and design of these profiling studies, relevant and robust analysis techniques are required. For instance, the use of unsupervised machine learning algorithms [16], such as clustering analysis [17,18,19], take into account the natural grouping structure of the subjects. This improves statistical accuracy/efficiency and allows the identification of patterns and subgroups useful for decision-making and planning interventions. Thus, this study aimed to determine the physical–functional profile of young Mexican boxers from Mexico City with >1 year of boxing experience through unsupervised machine learning on PIF, PV, maximal isometric handgrip strength (HG), and lower-limb muscle power data. This study will contribute to classifying male and female boxers aspiring to enter the Technological Baccalaureate of Education and Sports Promotion (BTED, in Spanish ‘Bachillerato Tecnológico de Educación y Promoción Deportiva’). It is hypothesized that the variation in the data could be explained mainly by strength levels and sex.

## 2. Materials and Methods

### 2.1. Study Design

A cross-sectional study of anthropometric data and physical–functional tests were performed and reported in accordance with the Strengthening the Reporting of Observational Studies in Epidemiology (STROBE) guidelines [20]. These guidelines are an international, collaborative initiative of epidemiologists, methodologists, statisticians, researchers, and journal editors involved in the conduct and dissemination of observational studies.

### 2.2. Setting

The study was conducted during the month of August 2021 with young boxers from Mexico City. The research was performed within the framework of the admission tests to enter the BTED (https://bachilleratodeportivo.sep.gob.mx, accessed on 16 November 2022). All the measurements of the young athletes were taken at a single moment in time at the G3 sports complex in the ‘Alcaldía Álvaro Obregón’ (Mexico City, Mexico). Evaluations were conducted between 9:00 a.m. and 5:00 p.m. (GMT-6) on the weekend.

### 2.3. Participants

A total of 227 young boxers performed the admission tests to enter the boxing Baccalaureate (high school) and were potentially eligible to participate in this study. These tests were previously approved by the school’s Educational Directorate. The sample contained Mexican amateur boxers at local, regional, and national levels. All procedures were developed in accordance with the latest version of the Declaration of Helsinki [21]. Informed signed consent was requested from the parents of each participant. After reviewing the informed consent, the participants were asked verbally if they agreed with the evaluation and were informed about their freedom to withdraw from the research at any time and without any repercussions. In the consent form, detailed information was given about the aim of the study, the measures to be performed, the approximate duration of the assessment, and the attending requirements (comfortable clothing).

### 2.4. Variables

The selected variables in this study were (i) demographic variables (body mass (BM, kg), sex, and age (years)); (ii) physical performance variables (time of flight in the countermovement jump (CMJ, s) and maximal isometric HG (kg)). The jump height of the CMJ was derived from the time of flight; and (iii) performance variables in boxing (PIF (kg) and the velocity of straight boxing punches (PV, m·s^−1^)). In addition, the following are reported: (i) the PIF-to-BM ratio, (ii) the PIF-to-HG ratio, and (iii) the PIF-to-BM+HG ratio.

### 2.5. Data Sources/Measurement

The measurements of the young athletes were carried out during the admission tests to enter the BTED. Each boxer went through the tests through three stations, divided into groups of 20 fighters. Before the assessment, the athletes performed a 10 min group warm-up involving specific boxing gestures. For better control and follow-up of the evaluation process, three measurement stations were placed on the left side of the gymnasium: (i) the first for the evaluation of BM and maximal isometric HG, (ii) the second to evaluate the CMJ, and (iii) the third to evaluate PV and PIF of straight boxing punches.

#### 2.5.1. Maximal Isometric Handgrip Strength (HG)

A digital hand-held dynamometer was used for this test (EH101, Camry Corp., El Monte, CA, USA). Participants were asked to perform a maximal isometric HG contraction for 5–6 s, as required to reach the peak rate of force development [22,23]. As recommended by several authors [24,25], two repetitions were requested whereas the rest time between repetitions was one minute [26]. The assessment of maximal isometric HG strength has been shown to be reliable (*r* between 0.85 and 0.99) [26]. An important use that can be made of this assessment is for injury prevention since it has been shown in several studies that isometric strength assessments provide predictive information for injuries associated with dynamic lifting tasks [27,28].

#### 2.5.2. Countermovement Jump (CMJ)

As an important component of the transmission of force during boxing strikes [29], we evaluated lower-limb muscle power with CMJ [30] following Bosco’s protocol [31]. The participant stood in an upright position on the contact mat with his eyes straight ahead and both hands on his hips. Upon hearing the signal emitted by the interface of the contact mat, the subject had to descend quickly with a knee flexion to an angle of approximately 90° (the trunk had to be as close as possible to the vertical axis), from that position the participant had to apply force with his lower limbs to rise quickly in a vertical direction as far as possible from the ground. The importance of keeping the lower limbs and trunk in full extension during the entire flight phase until landing on the carpet was emphasized. In the fall, the young boxer had to land with plantar flexion at the ankle and in knee and hip extension, in order to subsequently generate flexion of the articular nuclei and cushion the impact [32]. The CMJ height was estimated from the flight time calculated by a validated contact mat (WLAC02 v5.40, Win Laborat S.A, CABA, Argentina) [33]. The use of time-of-flight to estimate vertical jump height is not only a frequent strategy in exercise science [34] but also, in particular to CMJ, has shown good reliability with an Intraclass Correlation Coefficient of 0.97 (95% CI: 0.92–0.98) and a Coefficient of Variation of 2.6% [35].

#### 2.5.3. Punch Velocity (PV)

A boxer’s ability to rapidly activate the neuromuscular system, contract the active musculature and produce force at the highest possible velocity is an important characteristic of boxing performance [36]. To evaluate PV, the boxers were instructed to form a guard position behind a line on the ground, securing a standardized starting point. Participants were asked to perform as many punches in the air as possible within 6 s while holding the accelerometers in their hands. The test began when the athlete was asked if he/she was ready, and if he/she answered in the affirmative, he/she had to start the test at the “go” voice given by the evaluator. In the end, a record was taken of the maximum velocity achieved as indicated by the accelerometer application that communicates via Bluetooth with the researcher’s smartphone. The PV was evaluated using Corner Boxing Trackers v1.3.1 accelerometers (Corner Wearables Ltd., Manchester, UK). These devices have been shown to be valid by recording PV with a mean percentage error of 0.005, a mean absolute percentage error of 0.031, and a *p*-value of 0.014 in the two one-sided tests (test of equivalence) [37].

#### 2.5.4. Punch Impact Force (PIF)

The ability to hit with great force is beneficial to boxers because it generates a higher probability of success by influencing the judges’ perception of a clean hit, impairing the opponent’s fighting ability, and making knockouts possible [36,38]. In this study, the impact force of the straight rear hand boxing punch was measured. Previously, our research group developed and validated a methodology for assessing the force of the blow with a machine that contains a load cell (WLIT04 v5.40, Win Laborat S.A, CABA, Argentina) and a cushioning element [39]. The cell was placed inside a metallic structure, which has a shock-absorbing surface that reduces injury risk in the boxers’ fists [40] (Figure 1). To determine the impact force that the metal device with its damping elements (cushion and spring) can absorb, steel discs of 20, 40, 50, and 60 kg were placed on the load cell, and the data obtained by the device’s software were recorded. The data obtained were 0.88, 20.5, 35.2, and 45.5 kg, respectively. The dispersion of the data around the line of best fit showed a high precision of the measurement (*r* = 0.99).

The protocol to measure PIF in this study consisted of the following: (i) boxers were fitted with a 16-ounce glove on their dominant arm; (ii) were placed in a fighting stance (one foot forward and one foot back) with their arms in guard; (iii) the distance between the assessment device and the boxer was chosen individually considering the length of their arm, the rotation of their trunk and the impossibility of advancing with the arm or trunk to perform the punch [8]; (iv) at the sound signal coming from the load cell interface, young boxers performed three punches as hard as possible with their dominant hand (straight rear hand) with a three-second pause between each punch; (v) the impact with the highest force production was recorded. According to the manufacturer, the load cell takes data at one-millisecond intervals at a resolution of 12 bits, which is adequate for the variables to be measured in this work.

### 2.6. Sample Size

After the call to participate in the boxing sports baccalaureate, only the young boxers who submitted all relevant documentation and met all the criteria for admission were considered potentially eligible participants of this study (https://bachilleratodeportivo.sep.gob.mx/assets/convocatorias/CONVOCATORIA.pdf, accessed on 20 December 2022). Thus, a convenience non-probabilistic sample of 227 males and females was obtained.

### 2.7. Statistical Analysis

Descriptive statistics were expressed as the median and interquartile range (IQR). All data were analyzed with the Mann–Whitney U test to determine differences between sexes and between generated clusters considering the appropriateness, robustness, and flexibility of non-parametric statistics. The probability of superiority (ρ) was calculated as the effect size [41] using the formula ρ = U/(n_1_ × n_2_). In this formula, U represents the U statistic obtained from the Mann–Whitney U test, and n_1_ and n_2_ represent the sample sizes of the two groups being compared (i.e., female versus male [comparison between sexes]; profile 1 versus profile 2 [comparison between clusters]). A resulting value of ρ close to 0.5 suggests a more balanced comparison (no clear distinction in terms of superiority) between the groups [42]. As we have performed previously [43,44,45], the participants were subdivided into clusters using unsupervised machine learning to identify similar data points (natural groupings) and extract profile patterns. We used the partitioning around Medoids (PAM) algorithm, also known as k-Medoids clustering which unlike the k-means algorithm considers the median as the center of a cluster, thus, is more robust to noises and outliers. This has enhanced robustness against outliers and reduced noise in the unsupervised machine-learning process [46]. The number of clusters was determined a priori with the R package ‘NbClust’ by comparing the two-to-ten cluster solutions (frequency among all indices) of the following methods: “wss”, “silhouette”, “gap_start”, “kl”, “ch”, “hartigan”, “mcclain”, “gamma”, “gplus”, “tau”, “dunn”, “sdindex”, “sdbw”, “cindex”, “ball”, “ptbiserial”, “gap”, and “frey”. The internal validation for selecting the clustering method to discuss our results was performed with the ‘clValid’ package which implements an evaluation of the goodness of the resulting clusters by comparing Dunn, Silhouette and Connectivity criteria measures [47]. The package ‘factoextra’ was used to visualize the clustering results within the free software environment for statistical computing and graphics R v4.0.2 [48]. The statistical significance of the Mann Whitney U results was assessed using the IBM SPSS v26 (IBM Corp., Armonk, NY, USA), with a threshold of *p* < 0.05.

## 3. Results

### 3.1. Descriptive Data

All potentially eligible applicants to the BTED participated in this study. Thus, all data measurements were obtained from 227 young athletes with >3 years of experience in amateur boxing (Table 1).

Pearson correlations (*r*, 95% CI; *p*-value) were calculated to explore the relationship between variables. There were no significant correlations between PV and PIF-related variables. The correlations for the other variables can be seen in Figure 2 (full statistical results are reported in Appendix A). As is reported in the next sections, sex did not influence the profile generation.

### 3.2. Main Results

The k-Medoids clustering algorithm generated two profiles (clusters) of young boxers (Figure 3).

According to the distribution of female and male boxers in the Clusters (Cluster 1: F = 17, M = 101; Cluster 2: F = 27, M = 82), it seems that sex does not explain the variance of the data, although it must be considered that there were fewer women than men in the sample (44 vs. 183, respectively). Cluster 2 had the highest records on PIF, CMJ height, and PV. However, Cluster 1 had higher values of BM and maximum isometric HG strength (Table 2). After performing a data analysis per cluster, statistically significant differences (*p* < 0.001) with a large effect size (clear distinction in terms of superiority) were found on BM (ρ = 0.197), PIF (ρ = 0.118), the PIF-to-BM ratio (ρ = 0.017), the PIF-to-HG ratio (ρ = 0.079) and the PIF-to-BM+HG ratio (ρ = 0.008). It is important for readers to note that when ρ = 0.5, there is no effect. As the value diverges further from 0.5, the magnitude of the superiority increases.

## 4. Discussion

The weighting of specific physical variables could offer better parameters for the comparison of the physical–functional profiles of the fighters. In this study, we aimed to generate physical–functional profiles of young Mexican boxers using an unsupervised machine-learning algorithm. We identified two statistically significant different profiles.

Currently, the most widely used device in research to measure the punch impact force (PIF) is the piezoelectric force transducer embedded in the target to be hit [7,38,49]. Smith et al. reported significant differences between rear (mean [SD]; 2381 [328] N) and lead (1604 [273] N) hand punch force in novice boxers [38]. The authors also showed that the PIF depends on the boxer’s experience (being higher for elite boxers throwing straight rear hand punches) and, thereby, it can be used as a good parameter for athlete profiling. Another research group [8] also demonstrated higher PIF values for the super heavyweights (4345 [280] N) compared to the middleweight boxers (2625 [543] N). Contrary to what was expected given the higher BM in Cluster 1, the PIF was statistically different between the groups with a clear distinction in terms of superiority in favor of Cluster 2 (ρ = 0.118). In general, greater values of BM and maximal isometric HG strength were observed in Cluster 1 while Cluster 2 reported higher levels of PIF, CMJ height, and punch velocity (PV). The differences between clusters were more sensitive to variations in the strength (HG) and power (CMJ) variables. Interestingly, the clusters did not differ proportionally when matching participants by sex. Thus, we partially confirmed our initial hypothesis since cluster analysis revealed that muscle strength level but not sex could explain the variation in the data.

Recent studies have used physical tests to profile boxers or to identify the ranges in the specific variables (e.g., specific reaction speed) in accordance with boxing requirements [50,51,52]. Similar to our study, previous research has reported a high relationship between muscle strength and PIF in boxing [53,54,55]. Although we found significant differences between the sexes on HG and CMJ, which could be partially explained by the higher muscle development that males tend to show over females at these ages, the clustering algorithm resulted in sex-independent profiles. According to several authors, the maximal isometric HG is a valid indicator of upper limb strength [56] and therefore could represent a good parameter that is positively correlated with the ranking in competitive boxing [10]. Although a positive relationship has been established between maximal isometric HG strength and the power of the straight and the cross punch (*r* = 0.74 and *r* = 0.63, respectively) [57], our study did not find a significant correlation between PIF and HG. Furthermore, lower values of HG strength in young boxers are reported in this study compared to other findings in adults which possibly indicates an age-dependent phenomenon. In Mexican boxers (20.1 [2.16] years; 1.73 [0.10] m; 67.95 [16.73] kg), previous evidence showed mean HG strength values of 46.02 (9.56) and 43.89 (9.96) for the right and left arm, respectively [56]. Bruzas, Mockus, Cepulenas and Maciulis [57] reported maximal isometric HG values of 42 (12) kg and 65.5 (17.5) kg for the right and left arm, respectively, in competitive national Lithuanian boxers (22.5 [4.5] years; 1.77 [0.21] m; 79.75 [28.25] kg). On the other hand, Guidetti, Musulin, and Baldari [10] reported mean HG strength values of 58.2 (6.9) kg in Italian boxers (22.3 [1.5] years; 1.77 [0.2] m; 77.4 [1.4] kg). It is important to highlight that a large contribution to PIF derives from lower limb muscle strength, as it has been shown that increases in power measured through vertical jumping are related to an increase in PIF [58,59,60]. This agrees with our findings since the profile with better functional performance (Cluster 2) had the highest records on PIF, the CMJ height and PV.

The data obtained by performing a battery of tests to know the boxers’ profiles, allow for detecting important changes, as well as differentiating fighters of different categories and performance levels, since it is also known that the maturation process generates constant changes at physical, biological, and functional levels [61]. The term functional refers to sport-specific performance, such as the use of boxing techniques, punching techniques, and movement which can be affected by normal body growth, which occurs from infancy to adulthood [62]. However, being able to know which aspects seem to be more related to sports performance allows coaches and physical trainers to set more accurate objectives when programming training with these boxers. Most of the studies that characterize the physical sports profile are based on anthropometric indicators, physical capacity, or physiological parameters. These studies usually use approaches focused on descriptive analysis and correlations, mainly bivariate. Subsequently, comparisons are made with similar or reference populations, based on measures such as the mean or median. Although this approach provides valuable information, it is necessary to complement and deepen the characterization of athletes. It is worth noting that reported findings available in literature have been in male boxers, suggesting further research is warranted in amateur female boxers [2]. In fact, this is the first time that machine learning profiles are generated using physical–functional variables in both sexes, which might help in the monitoring process during the physical preparation of young boxers. In this sense, clustering analysis, using a set of multivariate statistical techniques, can help identify patterns of similarity and group performance variables to gain a better understanding and make more informed decisions in sports training. Thus, the use of cluster analysis could contribute to maximizing the quality and clinical relevance of the generated profiles, particularly in exercise and sports sciences.

### 4.1. Limitations and Future Directions

This study has several flaws that should be mentioned. First, there was a higher proportion of male athletes in the assessed participants. However, it should be noted that sex did not have statistical or clustering (topological) relevance in the generation of the profiles. Since the aim of this study was to classify a set of data (profiling) based on physical and boxing performance, future studies might focus on generating models to predict/estimate sports success or injury ratio including maturity status as a predictive variable. Second, we did not measure body composition variables beyond BM. Further research is needed to have a characterization that test associations between morphology (e.g., wingspan, upper and lower limb ratios), body composition (e.g., percent body fat and muscle mass), and boxing performance (e.g., CMJ, PV and PIF). Third, this research did not delve into substrate energy utilization (e.g., VO_2max_ and lactate). Regarding the results of the force profile applied in the boxing straight punch, care should be taken when comparing the data obtained in this study with that of other investigations, due to the substantial differences in the functioning of the equipment used to record this information: dynamometer [6,38]; accelerometer and load cell [8]; a cushioned mass suspended as a ballistic pendulum [7]; and video analysis with anatomical markers based on 12-segment models with a force platform [49]. Finally, after realizing the possible incidence of the level of technical management on the physical performance data obtained in this study, we believe that it would be convenient in future research to make a report on the technical mastery of boxers (e.g., won fights, knockouts) since it is likely that the specific coordination is responsible for some differences observed in the performance variables in boxing; or if possible, conduct a period of familiarization with the tests.

Besides providing information and distribution of the young boxing population, this study enriched outcome information through the analysis of physical–functional variables specific to boxing. The Research Division of DBSS International SAS-Mexico will soon initiate a project with the support of the Guanajuato Association of Mixed Martial Arts, to continue collecting quality data in this mission to enhance talent identification and derive/develop a method for the evaluation of the physical–functional profile in boxers and combat athletes.

### 4.2. Interpretation

This study provides relevant information to perform a physical–functional profile of young Mexican boxers, thus allowing a better understanding of the physical differences of athletes with sporting projection. In addition, it contributes to methodological procedures to analyze/interpret physical data sets under partially sport-specific biomechanics.

### 4.3. Generalizability

The relative lack of homogeneity of athletes by sex and BM does not allow extending the generalization to each geographic location in Mexico. It should be noted that to provide transparency in the selection of applicants for admission to the boxing sports baccalaureate, the evaluations should be objective, avoiding being affected by the subjectivity of the observer. To this end, the statistical treatment of the data obtained from the boxing-specific physical tests provides sufficient rigor in the determination of the clusters that will serve as a guide in the selection of applicants.

## 5. Conclusions

Using advanced statistical modeling techniques, we identified two physical–functional profiles of young Mexican applicants for admission to the baccalaureate in sports. In general, strength levels explained most of the variation in the data. Therefore, it is reasonable to recommend the implementation of tests aimed at assessing lower limb strength levels (CMJ) and applied strength (PIF and PV) in boxing gestures of aspiring high school athletic entrance candidates. The identification of these physical–functional profiles could help to differentiate training programs during the sports specialization of young boxers.

## Figures and Tables

**Figure 1 sports-11-00131-f001:**
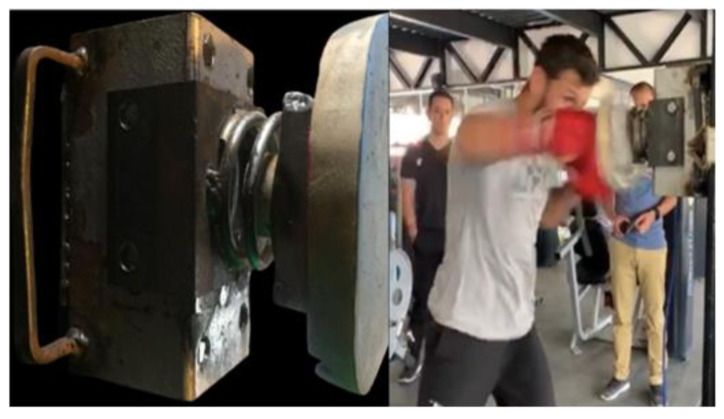
Measure of the straight rear hand punch impact force. Reproduced with permission from Merlo and Rodríguez-Chávez [39]. Source: the authors.

**Figure 2 sports-11-00131-f002:**
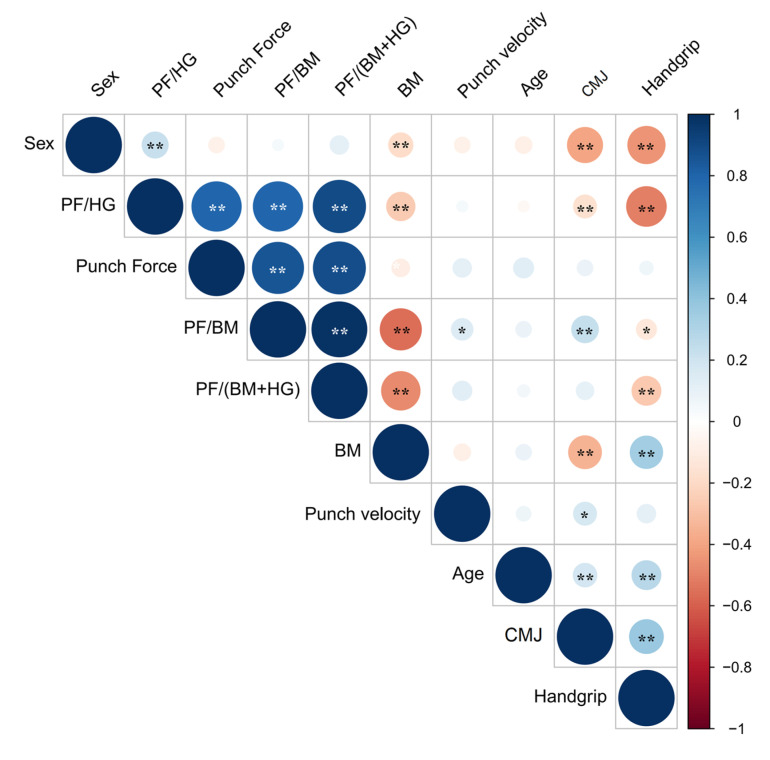
Draftsman correlation plot. Positive correlations are displayed in blue and negative correlations in red color. The color intensity and the size of the circle are proportional to the correlation coefficients. BM: body mass; CMJ: countermovement jump height; HG: maximal isometric handgrip; PF: punch impact force. * *p* < 0.05; ** *p* < 0.01.

**Figure 3 sports-11-00131-f003:**
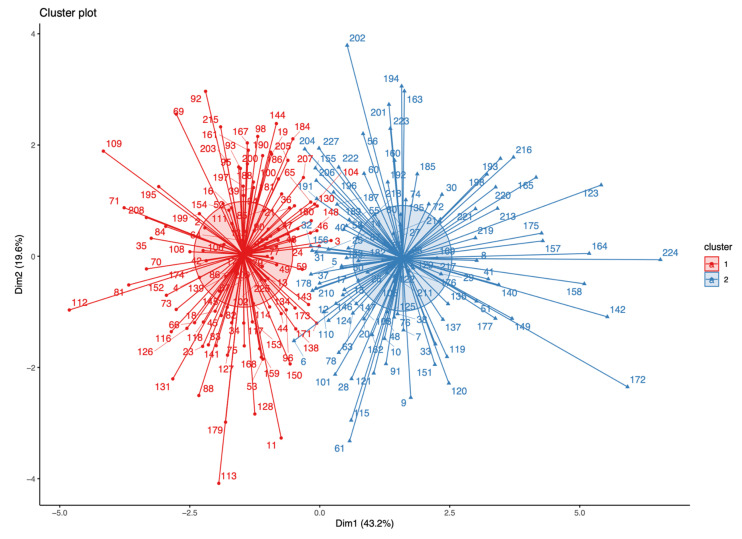
Cluster diagram of the k-Medoids analysis.

**Table 1 sports-11-00131-t001:** Characteristics of the participants by sex.

Variable	Female (n = 44)		Male (n = 183)		*p* Value	ρ
	Median	IQR (Q3 − Q1)	Median	IQR (Q3 − Q1)		
Age (years)	15	15.25 − 15	15	16 − 15	0.157	0.435
BM (kg)	57.80	63.00 − 49.95	60.80	73.10 − 53.55	0.004	0.360
CMJ (cm)	27.90	25.72 − 19.15	29.60	34.25 − 25.00	<0.001	0.193
HG (kg)	32.20	30.00 − 23.90	34.80	38.45 − 30.10	<0.001	0.153
PIF (kg)	57.63	63.14 − 42.75	54.42	69.61 − 43.98	0.288	0.448
PIF/BM	0.99	1.17 − 0.66	0.85	1.16 − 0.65	0.594	0.474
PIF/HG	1.77	2.77 − 1.62	1.61	1.98 − 1.24	0.001	0.344
PIF/(BM+HG)	0.63	0.78 − 0.48	0.56	0.72 − 0.43	0.107	0.421
PV (m·s^−1^)	8.30	9.77 − 6.40	8.40	10.20 – 6.80	0.459	0.464

Data are expressed as median and interquartile range (IQR). BM: body mass; CMJ: countermovement jump height; HG: maximal isometric handgrip; PIF: punch impact force; PV: punch velocity. The statistically significant differences at a level of 0.05 for the Mann–Whitney U test are shown. Effect size as the probability of superiority (ρ).

**Table 2 sports-11-00131-t002:** Characteristics of the generated profiles.

Variable	Profile 1 (*n* = 118)		Profile 2 (*n* = 109)		*p* Value	ρ
	Median	IQR (Q3 − Q1)	Median	IQR (Q3 − Q1)		
Sex (n, %)	17 F (14.4%); 101 M (85.5%)		27 F (24.7%); 82 M (75.2%)			
Age (years)	15	16 − 15	15	16 − 15	0.562	0.479
BM (kg)	65.9	80.67 − 59.20	53.9	59.80 − 50.20	<0.001	0.197
CMJ (cm)	26.6	32.20 − 22.92	29.00	33.30 − 24.60	0.041	0.421
HG (kg)	35.2	40.40 − 29.55	32.1	35.50 − 26.70	<0.001	0.362
PIF (kg)	45.27	50.02 − 37.80	67.75	76.81 − 57.63	<0.001	0.118
PIF/BM	0.66	0.78 − 0.54	1.17	1.42 − 1.01	<0.001	0.017
PIF/HG	1.32	1.55 − 1.06	2.03	2.72 − 1.77	<0.001	0.079
PIF/(BM+HG)	0.44	0.50 − 0.36	0.75	0.89 − 0.66	<0.001	0.008
PV (m·s^−1^)	7.65	9.67 − 6.40	8.70	10.60 − 7.30	0.010	0.401

Data are expressed as the median and interquartile range (IQR) unless otherwise indicated. BM: body mass; CMJ: countermovement jump height; F: female; HG: maximal isometric handgrip; M: male; PIF: punch impact force; PV: punch velocity. The statistically significant differences at a level of 0.05 for the Mann–Whitney U test are shown. Effect size as the probability of superiority (ρ).

## Data Availability

The data supporting the findings of this study are available from the corresponding author upon request.

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
