# Peer review of "Profiling the Physical Performance of Young Boxers with Unsupervised Machine Learning: A Cross-Sectional Study"

_sports, 2023, doi:10.3390/sports11070131_

Round 1
Reviewer 1 Report
I commend the authors on their thoughtful and comprehensive study of physical-functional profiles in young Mexican boxers. The use of unsupervised machine learning algorithms to identify patterns and subgroups within the data is a valuable contribution to the field.
- One point that could be further addressed is the authors' choice of statistical test for analyzing differences between groups. While the Mann-Whitney U test is a common non-parametric test that can be used to analyze ordinal or continuous data, readers would benefit from a more detailed justification for this choice over other commonly used parametric tests such as ANOVA or t-tests. Specifically, the authors could provide more insight into why their data is better suited for a non-parametric approach and what factors influenced their decision to use the Mann-Whitney U test (e.g. was it because certain assumptions were violated, was it due to the distribution, etc.).
- Lines 208 – 209 discuss comparing two-to-ten cluster solutions to choose the number of clusters. The readers will benefit from knowing what the criteria to select the two cluster was.
- Consider rewording line 213 (A significance level of p<0.05 was considered using the IBM SPSS> < .05 was considered using the IBM SPSS v26). Did you mean “The statistical significance of the Mann Whitney U results was assessed using IBM SPSS v26, with a threshold of p < .05.”
- Table 1 and 2: It’s uncommon (although not inherently incorrect) to report eta-squared with the Mann-Whitney U test. It may be more appropriate to use effect size measures that are specifically designed for non-parametric tests, such as the Hodges-Lehmann estimator, Cliff's delta, etc. That being said, eta squared can still be used, perhaps just provide a justification for the reader. Also, reporting means and standard deviations (M and SD) with Mann-Whitney U test is not common because this nonparametric test does not assume a normal distribution of the data, which makes the interpretation of mean and standard deviation less meaningful. However, it is possible to report the median and interquartile range (IQR) with Mann-Whitney U test, which provides a measure of central tendency and dispersion that is appropriate for non-normally distributed data.
- Line 226: Check language. I’m not 100% but I think , “IC del 95 %; valor p” is Spanish
- Figure 1: Should the diagonal of the corrplot (which shows the correlation between the same variable) be displaying a significant effect? If you’re using the corrplot package for this Figure, it might be better to set diag = F, so that the same variables aren’t shown in the diagonal.
Overall, the article presents an interesting and valuable contribution to the field of sports science, particularly in the area of athlete profiling using advanced statistical modeling techniques. The authors successfully identified two physical-functional profiles of young Mexican boxers and highlighted the importance of assessing lower limb strength and applied strength in boxing gestures when evaluating aspiring high school athletic entrance candidates. The article is well-structured and clearly presented, with a thorough explanation of the methods used and limitations of the study. However, some areas for improvement include the need for a more detailed justification for the choice of statistical tests used, a clearer explanation of the results, and more extensive discussion of the implications and future directions of the study. Overall, with some revisions, this article has the potential to make a significant contribution to the field of sports science and athlete profiling.
Mostly fine, I only detected one line that is potentially in Spanish
Reviewer 2 Report
Reconsider after Major Revisions: The acceptance of the manuscript depends on the revisions. The author must provide a rebuttal if some comments cannot be revised.
Errores:
Line 178 - Figure 1. Measure of the straight rear hand punch impact force. Taken from [39].
Lines (232-235) - Figure 1. Draftsman correlation plot. Positive correlations are displayed in blue and negative correlations in red color. The color intensity and the size of the circle are proportional to the correlation coefficients. BM: body mass; CMJ: countermovement jump height; HG: maximal isometric handgrip; PF: punch impact force. * P < 0.05; ** P < 0.01.
Introduction:
It is well structured.
Methodology:
I d´not agree with the sample being with both groups (male + female).
Although, strangely, the values, for example, of the HG are little different for men than for women, I nevertheless believe that the Results should be presented by sex.
The Discussion:
Should follow the same guidelines and not discuss study values (male and female together) with literature values for males only, or young ages with adult ages, which happens in the Discussion chapter.
Conclusions:
They can also be quite different when the results and discussion can be carried out separately.
Round 2
Reviewer 2 Report
There are two figures with # 1. They must be changed.